# Defect Surface Engineering of Hollow NiCo_2_S_4_ Nanoprisms towards Performance-Enhanced Non-Enzymatic Glucose Oxidation

**DOI:** 10.3390/bios12100823

**Published:** 2022-10-04

**Authors:** Xiaomin Lang, Dandan Chu, Yan Wang, Danhua Ge, Xiaojun Chen

**Affiliations:** 1College of Chemistry and Molecular Engineering, Nanjing Tech University, Nanjing 211800, China; 2Jiangsu Key Laboratory of Molecular Biology for Skin Diseases and STIs, Nanjing 210042, China

**Keywords:** phosphorus doping, NiCo_2_S_4_, non-enzymatic sensor, electrochemistry

## Abstract

Transition metal sulfides have been explored as electrode materials for non-enzymatic detection. In this work, we investigated the effects of phosphorus doping on the electrochemical performances of NiCo_2_S_4_ electrodes (P-NiCo_2_S_4_) towards glucose oxidation. The fabricated non-enzymatic biosensor displayed better sensing performances than pristine NiCo_2_S_4_, with a good sensitivity of 250 µA mM^−1^ cm^−2^, a low detection limit (LOD) of 0.46 µM (S/N = 3), a wide linear range of 0.001 to 5.2 mM, and high selectivity. Moreover, P-NiCo_2_S_4_ demonstrated its feasibility for glucose determination for practical sample testing. This is due to the fact that the synergetic effects between Ni and Co species, and the partial substitution of S vacancies with P can help to increase electronic conductivity, enrich binary electroactive sites, and facilitate surface electroactivity. Thus, it is found that the incorporation of dopants into NiCo_2_S_4_ is an effective strategy to improve the electrochemical activity of host materials.

## 1. Introduction

The accurate measurement of glucose concentration from blood or other sources is of great significance to diagnose diabetes and monitor food quality in the field of pharmaceuticals and foods [1]. Recently, electrochemical sensors provide such simple operation, rapid response, and high sensitivity that they have aroused wide attention for use in glucose detection. Commercial glucose biosensors are generally based on the glucose oxidase enzyme (GO_x_), where glucose is converted into gluconolactone in the presence of saturated oxygen. The enzymatic sensors exhibit outstanding sensitivity and selectivity but suffer from several limitations caused by environmental effects, high cost, and tedious enzyme immobilization process [2]. In order to break the above-mentioned drawbacks, it is essential to develop and explore the non-enzymatic sensors for direct electrooxidation of glucose. Over the past decades, thanks to their highly active area, excellent stability, and relatively cheap price, various transition metal chalcogenides, including oxides, sulfides, and selenides, have been of great promise in electrochemical sensors. Among these, transition metal sulfides with high electrochemical activity have drawn extensive attention [3,4,5,6]. Compared with oxides, sulfides have a more striking electrochemical performance with outstanding electronic conductivity and abundant redox chemical properties, because of their more flexible structure with an elongation of chemical bonds constructed by replacing O with S, which benefits electron transport [7,8]. In particular, NiCo_2_S_4_ usually exhibits high electrochemical activities as a kind of single-phase binary metal sulfide, rather than the simple mixture of NiS_x_ and CoS_x_, due to rich chemical redox, low cost, complex chemical compositions, abundant resources, environmental friendliness, and the synergetic effect of both individual components [9]. In spite of promising results, NiCo_2_S_4_ still suffers from a reduced density of electrochemically active sites and severe polarization on account of lower conductivity and volume changes during electrochemical reactions. It still cannot meet the demands of high capacitance, which brings a challenge to its practical utilization [10,11]. Therefore, enormous research efforts have been devoted to enhancing the electrochemical performance of NiCo_2_S_4_ through regulating nanostructures or combinations with conductive materials.

Typical nanostructures of nanowires, nanotubes, or nanorods could remarkably enlarge the accessible areas between the electrolyte and electrode and shorten the ionic diffusion distance, while introducing conductive materials may optimize the electrical conductivity and extend the electroactive surface [3,12,13]. For example, Huang’s group fabricated a non-enzymatic glucose sensor based on the 3D flower-like NiCo_2_S_4_ deposited Ni-modified cellulose filter paper, exhibiting a wide linear range of 0.5 μM–6 mM, high sensitivity of 283 μA mM^−1^ cm^−2^, and a low detection limit (LOD) of 50 nM [14]. Guo et al. reported a NiCo_2_S_4_ nanowire array with a unique core-shell structure grown on electrospun graphitic nanofiber film, endowed with a wide linear range and a low LOD for glucose sensing [15]. Although NiCo_2_S_4_ could achieve outstanding electrochemical properties through the above two methods, it is not always feasible to modulate the inherent electronic properties to ideal levels. Recent reports have demonstrated that an additional surface-modified coating can change surface charge, microenvironment, and active site exposure, to achieve the purpose of adjusting activity [16]. Especially, anionic doping is verified as an effective method to introduce the surface defects for altering electron density, thereby improving the redox reactivity of the NiCo_2_S_4_ complex [13,17,18]. Among them, the phosphorus (P) element with fully vacant 3D orbitals and lone-pair electrons has drawn enormous attention for promoting the surface electroactivity of host materials. Thereby, the surface P-anion doping of NiCo_2_S_4_ might accommodate the surface charge state by tuning the partial charge density of neighboring bonded Ni and Co atoms. This method would enhance the redox activity of NiCo_2_S_4_ by optimizing the adsorption energy of electrochemical analytes and reducing the strain during redox reactions [13,19,20,21].

Inspired by these analyses, P-doping NiCo_2_S_4_ with a hollow nanoprism-like structure (P-NiCo_2_S_4_ HNPs) is successfully synthesized in this work through a simple phosphatization reaction. Benefiting from the unique hollow structure and surface engineering of NiCo_2_S_4_, the P-NiCo_2_S_4_ HNPs have offered a better electrochemical sensing performance than that of pristine NiCo_2_S_4_, such as higher sensitivity, better selectivity, and reproducibility, as well as excellent feasibility toward glucose determination in practical samples.

## 2. Materials and Methods

### 2.1. Chemicals and Materials

All reagents are of analytical grade and used without further purification. Double distilled water was used throughout the experiments. Cobalt (II) acetate tetrahydrate (Co(OAc)_2_·4H_2_O), nickel (II) acetate tetrahydrate (Ni(Oac)_2_·4H_2_O), potassium hexacyanoferrate (II) (K_4_[Fe(CN)_6_]), potassium ferricyanide (K_3_[Fe(CN)_6_]), D-(+)-glucose, thioacetamide (TAA), uric acid (UA), and sodium hypophosphite (NaH_2_PO_2_) were purchased from Sinopharm Chemical Reagent Co., Ltd. (Shanghai, China). Ascorbic acid (AA), ethanol (EtOH), potassium chloride (KCl), and 4-acetamidophenol (AP) were ordered from Aladdin Industrial Co., Ltd. (Shanghai, China), Yasheng Chemical Co., Ltd. (Jiangsu, China), Lingfeng Chemical Reagent Co., Ltd. (Shanghai, China), and Shanghai Macklin Biochemical Co., Ltd. (Shanghai, China), respectively. Urea and sodium chloride (NaCl) were acquired from Xilong Chemical Co., Ltd. (Guangdong, China), while L-cysteine (Lcy) and D-fructose (Fru) were obtained from Huixing Biochemical Reagent Co., Ltd. (Shanghai, China).

### 2.2. Apparatus

The morphologies and structures of the as-prepared samples were characterized by using scanning electron microscopy (SEM, Zelss Sigma300) at 3 kV and transmission electron microscopy (TEM, FEI Talos-F200s) at a voltage of 200 kV. The crystalline structure and chemical states of the samples were verified by powder X-ray diffraction (XRD, D/max-2500 diffractometer, Rigaku, Japan), energy-dispersive spectroscopy (EDS), and X-ray photoelectron spectroscopy (XPS, Kratos Axis Ultra DLD spectrometer). Surface area allowing Brunauer-Emmett-Teller (BET) isotherms was carried out by monitoring N_2_ adsorption/desorption using a NOVA 2000 surface area analyzer (Quantachrome) at 77 K.

### 2.3. Fabrication of the P-NiCo_2_S_4_ HNPs

According to the previous literature, NiCo_2_S_4_ NPs were firstly synthesized [22]. Briefly, Co(OAc)_2_·4H_2_O (0.25 g), Ni(OAc)_2_·4H_2_O (0.12 g), and urea (0.56 g) were successively added into EtOH (80 mL) under stirring. Then, the resulting mixture was heated to 65 °C for 4 h, and Co/Ni precursors were obtained after processing. Then, the Co/Ni precursors were re-dispersed into EtOH, followed by the addition of TAA. Subsequently, the resulting solution was transferred into a Teflon-lined stainless-steel autoclave and heated to 160 °C for 6 h. NiCo_2_S_4_ NPs were obtained after processing and drying in the air. Finally, NiCo_2_S_4_ (0.05 g) and NaH_2_PO_2_ (0.2 g) were placed in tandem and annealed at 350 °C for 2 h under a flowing N_2_ atmosphere to produce the P-NiCo_2_S_4_ HNPs.

### 2.4. Electrochemical Measurements

Before preparing modified electrodes, the indium tin oxide (ITO, diameter of 3 mm) electrodes were successively sonicated and cleaned with acetone, EtOH, and water for use. After sonication for an adequate time, 6 µL of a uniform dispersion of P-NiCo_2_S_4_ HNPs in EtOH (1 mg mL^−1^) was drop-casted onto the prepared ITO and dried at room temperature for next use. Cyclic voltammogram (CV) and chronoamperometry were performed by a CHI 660D instrument (Chenhua Instrument Co., Shanghai, China) to evaluate the electrochemical performance of the samples. In addition, the glucose detection was carried out in 0.2 M NaOH solution.

## 3. Results and Discussion

As illustrated in Figure 1A, the NiCo_2_S_4_ was synthesized using our previous method [22], and the synthetic procedure for P-NiCo_2_S_4_ was described through a facile P-doping. Utilizing CO_3_^2−^ and OH^−^ derived from the hydrolysis of urea in the presence of Co^2+^ and Ni^2+^, the Co/Ni precursors were obtained by a solution reaction and then employed as the self-engaged templates. The XRD pattern confirmed that the crystalline Co/Ni precursors possessed a tetragonal cobalt/nickel acetate hydroxide phase (Appendix A) [22]. Moreover, The EDS spectrum substantiated the successful synthesis of Co/Ni precursors with the mole ratio of Co to Ni of 2:1 (Appendix A). In addition, the morphologies and structures were further characterized by SEM. As shown in Figure 1B,C, the uniform prism-like Co/Ni precursors possessed a length of ~1.3 µm and a width of ~280 nm with a smooth surface. After sulfidation and the follow-up phosphating process, the length and width of the resulting P-NiCo_2_S_4_ were ~1.5 µm and ~250 nm, respectively (Figure 1D,E). It could be seen that the P-NiCo_2_S_4_ HNPs still maintained the original prism-like appearance with a hollow structure and rough surface, which was of benefit to enhancing the electron transfer between interfaces. As shown in Figure 1F of the TEM image of the P-NiCo_2_S_4_ HNPs, this further demonstrated the detailed geometrical morphology and the hollow interior structure. The XRD pattern of pristine P-NiCo_2_S_4_ was presented in Figure 2A. Obviously, all the diffraction peaks can well correspond to the cubic NiCo_2_S_4_ phase (JCPDS Card. No. 20-0782) [13,22,23]. The result indicated that the NiCo_2_S_4_ crystal structure was not distorted significantly by the introduction of P [13,23]. In accordance with the XRD analysis, the high-resolution TEM (HRTEM) depicted that the inter-planar distances of P-NiCo_2_S_4_ crystals were ~0.28 nm and ~0.56 nm, corresponding to the (311) and (111) planes of P-NiCo_2_S_4_, respectively (Figure 1G). Additionally, the corresponding EDS mapping further confirmed the uniform distribution of Co, Ni, S, and P elements within the hollow prisms (Figure 1H). Such results stayed in step with the EDS spectrum (Appendix A).

As shown in Figure 2, XPS was performed to further identify the chemical environment and surface element state of the as-prepared P-NiCo_2_S_4_ HNPs. The XPS survey spectrum confirmed the presence of P, S, O, Co, and Ni elements on the surface of P-NiCo_2_S_4_ HNPs (Figure 2B). For the 2p spectrum of Co (Figure 2C), it could be noticed that two main peaks at 782.4 and 798.7 eV were consistent with Co 2p_3/2_ and Co 2p_1/2_ of Co^2+^. Two other peaks at 778.8 and 793.9 eV could be indexed to Co 2p_3/2_ and Co 2p_1/2_ of Co^3+^, respectively, together with two satellite peaks at 786.2 and 803.5 eV [5,24]. The Ni 2p spectrum exhibited similar characteristics, as shown in Figure 2D. Two peaks at 853.3 and 870.2 eV were attributed to Ni 2p_3/2_ and Ni 2p_1/2_ of Ni^2+^, while two additional peaks at 857.4 and 871.3 eV resulted from Ni 2p_3/2_ and Ni 2p_1/2_ of Ni^3+^, respectively [25]. For the 2p spectrum of the S element (Figure 2E), the peak at 161.7 eV (S 2p_3/2_) was ascribed to the metal-sulfur bonds, while the binding energy of 162.8 eV (S 2p_1/2_) was linked to S^2−^ with low coordination at the surface. In addition, the two signals at 129.6 and 130.2 eV were assigned to P 2p_3/2_ and P 2p_1/2_, relating to metal phosphides. Moreover, the peak at 133.4 eV belonged to the oxidized phosphorus species (PO_x_), which originated from the exposure of its surface to air for ages [26]. Finally, the XPS result suggested a P content of about 3.8 at%. Therefore, the XPS results demonstrated the doping of P in NiCo_2_S_4_, which would increase the number of active sites, improve its conductivity, and thus enhance its electrochemical performance.

The CV was conducted to investigate the mechanism and electrochemical sensor activity of the P-NiCo_2_S_4_ HNPs modified electrode for glucose oxidation via a standard three-electrode system in a 0.2 M NaOH solution at a scan rate of 50 mV s^−1^. As shown in Figure 3A, although the CVs of bare ITO in the absence (black line) and presence (red line) of 1 mM glucose in an applied potential range of 0–0.6 V were active for glucose electrooxidation, the response was very weak. Figure 3B displays the CV curves of P-NiCo_2_S_4_/ITO with cathodic and anodic peaks at about 0.4 and 0.45 V, respectively. It was noticeable that a higher glucose oxidation peak was observed when adding 1 mM glucose, suggesting the excellent electrocatalytic activity of P-NiCo_2_S_4_/ITO towards glucose oxidation. The corresponding reaction mechanism for the glucose oxidation on the P-NiCo_2_S_4_ HNPs modified electrode can be briefly described as follows:CoS + OH^−^ ⟷ CoSOH + e^−^(1)
CoSOH + OH^−^ ⟷ CoSO + H_2_O + e^−^
(2)
NiS + OH^−^⟷ NiSOH + e^−^(3)

In addition, it can be easily seen from a comparison of current increments (ΔI) for bare ITO and P-NiCo_2_S_4_/ITO in Figure 3C that the electrochemical performance of the electrode modified with P-NiCo_2_S_4_ was greatly improved. Figure 3D presents the CVs of P-NiCo_2_S_4_/ITO with different glucose concentrations to evaluate the feasibility of glucose sensing, and the anodic peak current significantly increased with increasing concentration from 0 to 10 mM, representing typical catalytic oxidation of glucose. As shown in Figure 3E, the CVs of P-NiCo_2_S_4_/ITO at a series of scan rates with 1 mM glucose suggested that increasing the scan rate will lead to a rise in currents and a shift in potentials, which mainly comes from the increasing internal diffusion resistance within P-NiCo_2_S_4_ HNPs as the scan rate increases [2,27]. Furthermore, Figure 3F exhibits a good linear dependence relation between the anodic peak current and the square root of the scan rate with a linear regression coefficient (R^2^) of 0.9838, implying that the glucose oxidation on the P-NiCo_2_S_4_/ITO is a reversible and typical diffusion-controlled electrochemical process [28].

The identification of the optimal concentration of NaOH solution for an effective non-enzymatic glucose sensor was monitored at P-NiCo_2_S_4_/ITO for a range of 0.01~0.5 M (Figure 4A). Upon the increment in concentration from 0.01 to 0.2 M, the electrooxidation kinetics of glucose was dramatically enhanced, and the utmost response towards glucose was observed in 0.2 M NaOH solution. High-alkaline conditions will generate higher oxidation state species (Ni^2+^/Ni^3+^ and Co^3+^/Co^4+^), which provide the maximal glucose oxidation responses at P-NiCo_2_S_4_/ITO. However, stronger alkaline conditions tend to corrode electrodes resulting in limiting the electrochemical reaction [14]. Next, the optimal value of the applied potential was achieved in order to investigate the effect of detection potential on the amperometric response of P-NiCo_2_S_4_/ITO to glucose. As shown in Figure 4B, the oxidation current increased with increasing the detection potential and reached the highest value at 0.4 V. Therefore, 0.4 V was chosen as the optimal working potential for the follow-up study. Subsequently, for a systematic comparison, the P-NiCo_2_S_4_ nanoprisms with different amounts of NaH_2_PO_2_ were also prepared under the same synthetic conditions. In addition, the amperometric current responses of P-NiCo_2_S_4_/ITO with 0.1 g, 0.2 g, and 0.4 g NaH_2_PO_2_ at 0.4 V were represented in Figure 4C, which exhibited an almost obvious current difference. It was found that 0.2 g NaH_2_PO_2_ was the optimal admixing quantity in the following experiments. To analyze the interfacial behaviors of P-NiCo_2_S_4_ with 0.1 g, 0.2 g, and 0.4 g NaH_2_PO_2_, the electrochemical impedance spectroscopy (EIS) was performed in 0.1 M KCl solution containing 5 mM K_3_[Fe(CN)_6_]/K_4_[Fe(CN)_6_]. Obviously, the P-NiCo_2_S_4_ with 0.2 g NaH_2_PO_2_ showed a lower charge-transfer resistance (diameter of the semicircle) value than those of the other two, revealing the highest electrical conductivity of P-NiCo_2_S_4_ with 0.2 g NaH_2_PO_2_, and the result was consistent with Figure 4C. P-doping content has an effect on the electrical conductivity and electrocatalytic activity of materials. The high content of P-doping can significantly improve the fixation site and catalytic site of polysulfide, and thus promote the catalytic activity of materials. However, excessive P-doping will seriously affect the structural integrity of the materials, which significantly reduces the electrical conductivity [19,29,30]. The appropriate amount of P-doping can accelerate electron transport, and thus it would be a promising electrode to fabricate sensitive sensors. As shown in Appendix A, the BET surface areas of P-NiCo_2_S_4_ with 0.2 g and 0.4 g NaH_2_PO_2_ were 53.679 and 56.725 m^2^ g^−1^, respectively, which were slightly higher than that of P-NiCo_2_S_4_ with 0.1 g NaH_2_PO_2_ (43.558 m^2^ g^−1^). However, compared with pure NiCo_2_S_4_, P-doping can improve the specific surface area and electrochemical activity. Besides, the effective surface area of working electrodes will directly influence their sensitivity, thus playing a large role in developing electrochemical sensors. The CVs of P-NiCo_2_S_4_/ITO were measured in 5 mM K_3_Fe(CN)_6_ containing 0.1 M KCl solution, shown in Appendix A. Additionally, Appendix A indicates that the I_p_ values of P-NiCo_2_S_4_/ITO increase linearly with the square root of scan rates. Therefore, the effective surface area of the P-NiCo_2_S_4_ was assessed by the Randles–Sevcik equation: I_p_ = (2.69 × 10^5^)n^3/2^AD^1/2^v^1/2^C, where n was the number of electrons transferred, D was the diffusion coefficient of Fe(CN)_6_^3−^ (7.6 × 10^−6^ cm^2^ s^−1^), C was the reactant concentration (mol cm^−3^), v is the scan rate (V s^−1^), A was the effective area of the electrode, and I_p_ was the peak current [31,32]. The slope was the ratio of I_p_ to v^1/2^ (Appendix A); thus, the effective surface area of the P-NiCo_2_S_4_ was calculated as 0.134 cm^2^.

The amperometric analysis was used as an extremely attractive electrochemical technique to study the sensitivity, selectivity, and detection limit of the proposed glucose sensor. Under the above-optimized conditions, Figure 5A depicts a typical current-time plot of the P-NiCo_2_S_4_/ITO on the successive step-wise addition of glucose. The inset in Figure 5A represents the magnification of glucose concentration ranging from 1 to 4 µM. A wide linear response of P-NiCo_2_S_4_/ITO for glucose sensing from 0.001 to 5.2 mM was shown in Figure 5B. The linear regression equation for P-NiCo_2_S_4_/ITO was I/μA = 17.66C/mM + 1.72 (R^2^ = 0.9903) with a high sensitivity of 250.0 μA mM^−1^ cm^−2^. As a consequence, the LOD of the sensor was calculated to be 0.46 μM (S/N = 3). In addition, after adding glucose to the NaOH solution, the P-NiCo_2_S_4_ sensor produced steady-state signals less than 0.1s (Figure 5C). Table 1 lists the detection performance comparison of P-NiCo_2_S_4_/ITO with some similar non-enzymatic glucose sensors. It is clear that the performance of P-NiCo_2_S_4_/ITO is comparable or superior to some reports, for example, Co_3_O_4_/NiCo_2_O_4_, NiCo_2_S_4_/Pt, and CoNi_2_S_4_@NCF [33,34,35]. The remarkable electrochemical performance of P-NiCo_2_S_4_/ITO could stem from the hollow structures, synergetic effects between Ni and Co species, and partial substitution of S vacancies with P. This can provide more electrochemical active sites and enhance electrical conductivity, implying the potential value of P-NiCo_2_S_4_ in many systems for glucose detection.

Selectivity and stability are the major factors to evaluate the performance of the non-enzymatic glucose sensor. It is clear that the P-NiCo_2_S_4_/ITO displays an anti-interference advantage after the addition of glucose (1 mM) and a series of interfering species (0.1 mM each) at a working potential of 0.4 V, shown in Figure 5D, indicating the good ability of anti-interference of P-NiCo_2_S_4_ with negligible interference from urea, NaCl, KCl, AA, UA, Fru, Lcy, and AP. The long-term stability of the P-NiCo_2_S_4_/ITO electrode was evaluated by measuring the current peak towards 1 mM glucose over a week at room temperature. The electrode basically kept the same value as the original current after 7 days, showing good stability of the P-NiCo_2_S_4_ electrode at room temperature (Figure 6). The enhanced electrochemical behavior may be attributed to: (i) more redox reactions from multiple oxide states of the Ni and Co species, (ii) a much lower optical band gap energy and higher electric conductivity of binary metal sulfides, (iii) its unique hollow structure that enlarges surface area and shortens electronic transmission, and (iv) the anionic phosphorus doping that provides some new active sites and introduces surface defects.

In addition, the feasibility of the proposed electrode was evaluated by monitoring glucose in human serum. The serum samples were donated by Jiangsu Province Hospital. The serum sample was firstly separated by centrifugation at 8000 rpm for 15 min, then the collected supernatant was obtained for further electrochemical tests. The standard addition method was employed to detect the glucose concentration in serum under optimal experimental conditions. The successive addition of glucose was continuously added into the mixture of 9.5 mL of 0.2 M NaOH and 0.5 mL of the treated serum supernatant (Appendix A). The corresponding calibration curve of glucose ranging from 0.005 to 5.2 mM on the P-NiCo_2_S_4_/ITO was illustrated in Appendix A, and the linear regression equation was ΔI/μA = 4.62C/mM + 36.51 (R^2^ = 0.9891), disclosing good stability and anti-interference of the proposed glucose sensor for real blood samples. Moreover, we also measured the recoveries in serum samples by adding glucose solution with different concentrations. As presented in Table 2, the recoveries were calculated to be in the range of 96.4–101.8%. The above results demonstrated that the P-NiCo_2_S_4_/ITO had favorable feasibility of glucose detection in real serum samples.

## 4. Conclusions

In summary, we have rationally synthesized hollow NiCo_2_S_4_ nanoprisms in situ using a urea precursor as a sacrifice template by an anion exchange reaction, followed by constructing P-NiCo_2_S_4_ through simple P element doping for enhancing the electrochemical performance towards glucose oxidation. Compared with pristine NiCo_2_S_4_, a non-enzymatic sensor with good sensitivity, low detection limit, wide linear range, and high selectivity was established. It is confirmed that the incorporation of P can induce the surface defect at the atomic level to improve the electrochemical activity of host materials, which often suffer from slow ion diffusion and charge transfer kinetics, and the deficiency of electrochemically active sites. A series of electrodes with different amounts of P suggest that an appropriate amount of P-doping can accelerate electron transport. Moreover, P-NiCo_2_S_4_ demonstrates its feasibility for glucose determination for practical sample testing. These results reveal that the synergetic effects between Ni and Co species and the partial substitution of S vacancies with P can help to increase electronic conductivity, enrich binary electroactive sites, and boost surface electroactivity. Thus, our proposed non-enzymatic sensor has great prospects in the application of detecting glucose rapidly and effectively. In addition, the reported nanomaterials-based electrodes have been mostly applied in strongly alkaline conditions for non-enzymatic glucose sensors so far. However, the electrochemical characteristics in weakly alkaline or even neutral environments close to human body fluid are rarely reported. It is possible that, before the test, a negative potential is first employed to reduce the water in situ, and the hydrogen gets released resulting in the formation of OH^−^; thus, it produces an alkaline microenvironment, so as to realize glucose detection in a neutral environment. This will be further studied in the future.

## Figures and Tables

**Figure 1 biosensors-12-00823-f001:**
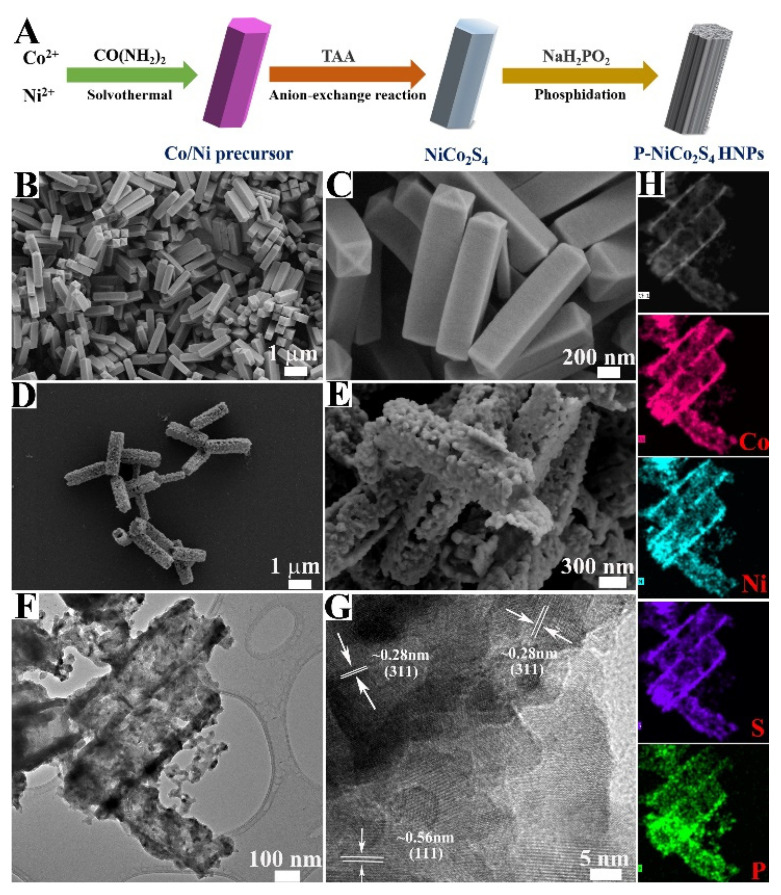
(**A**) Schematic illustration of P-NiCo_2_S_4_ HNPs preparation. SEM images of: (**B**,**C**) Co/Ni precursor, and (**D**,**E**) P-NiCo_2_S_4_. (**F**) TEM, (**G**) HRTEM image, and (**H**) EDS mapping of P-NiCo_2_S_4_ HNPs.

**Figure 2 biosensors-12-00823-f002:**
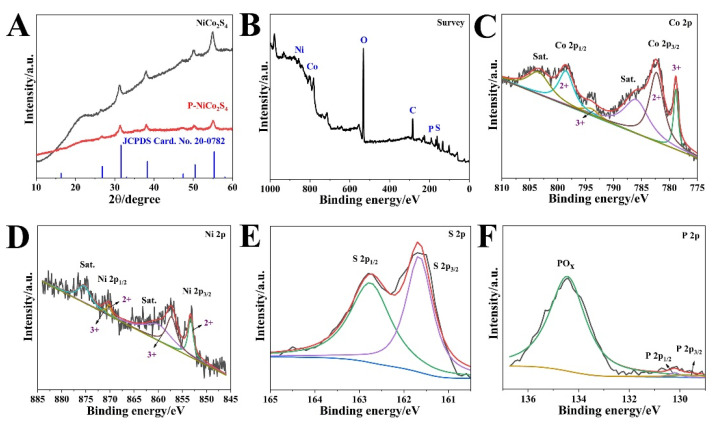
(**A**) XRD patterns of P-NiCo_2_S_4_ and NiCo_2_S_4_. (**B**) Full range XPS survey spectrum. The 2p spectrum of: (**C**) Co, (**D**) Ni, (**E**) S, and (**F**) P elements in P-NiCo_2_S_4_ HNPs.

**Figure 3 biosensors-12-00823-f003:**
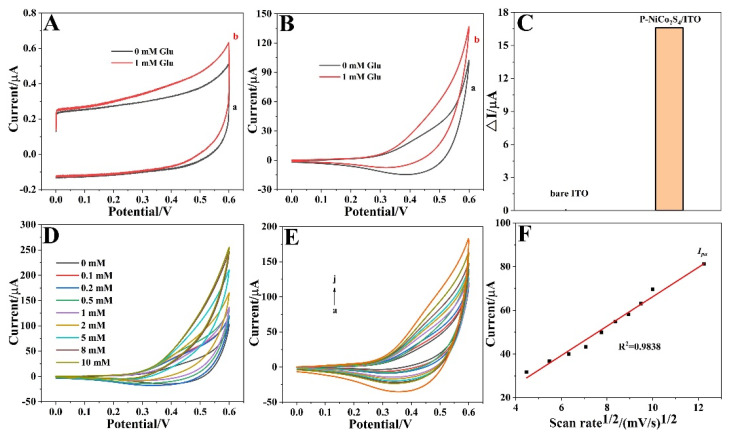
CVs of: (**A**) bare ITO, and (**B**) P-NiCo_2_S_4_/ITO in 0.2 M NaOH without (a) and with (b) 1 mM glucose (scan rate: 50 mV s^−1^). (**C**) A bar chart for comparison of current increments (ΔI) for bare ITO and P-NiCo_2_S_4_/ITO. (**D**) CVs of P-NiCo_2_S_4_/ITO in 0.2 M NaOH in the presence of different glucose concentrations at a scan rate of 50 mV s^−1^. (**E**) CVs of P-NiCo_2_S_4_/ITO in 0.2 M NaOH with 1 mM glucose at different scan rates (a to j: 20, 30, 40, 50, 60, 70, 80, 90, 100, and 150 mV s^−1^). (**F**) The corresponding plot of anodic current density (I_pa_) vs. the square root of the scan rate.

**Figure 4 biosensors-12-00823-f004:**
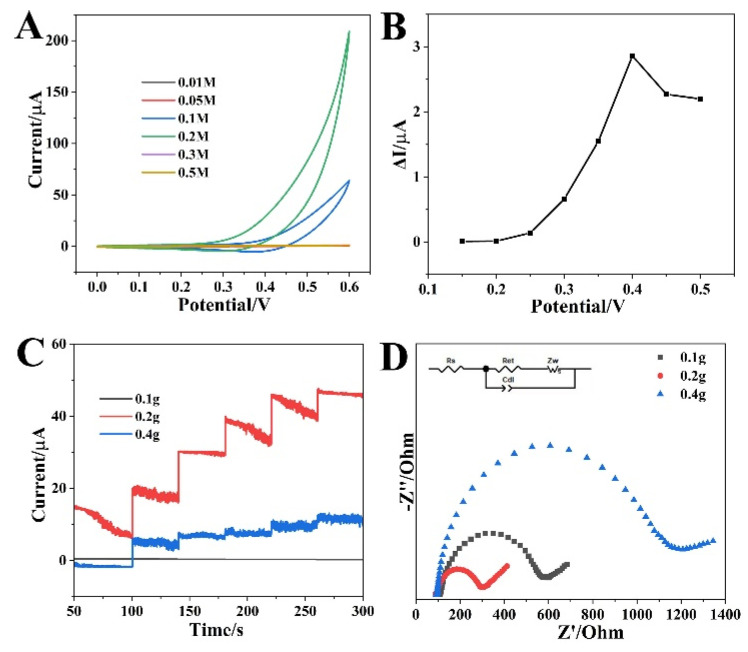
(**A**) CVs of P-NiCo_2_S_4_/ITO in different concentrations of NaOH solution with 1 mM glucose (scan rate: 50 mV s^−1^). (**B**) Current responses of P-NiCo_2_S_4_/ITO at different detection potentials with 1 mM glucose in 0.2 M NaOH. (**C**) Amperometric responses of P-NiCo_2_S_4_/ITO with different amounts of NaH_2_PO_2_ at 0.40 V with continuous addition of 1 mM glucose in 0.2 M NaOH. (**D**) Nyquist plots for P-NiCo_2_S_4_ electrodes with different amounts of NaH_2_PO_2_.

**Figure 5 biosensors-12-00823-f005:**
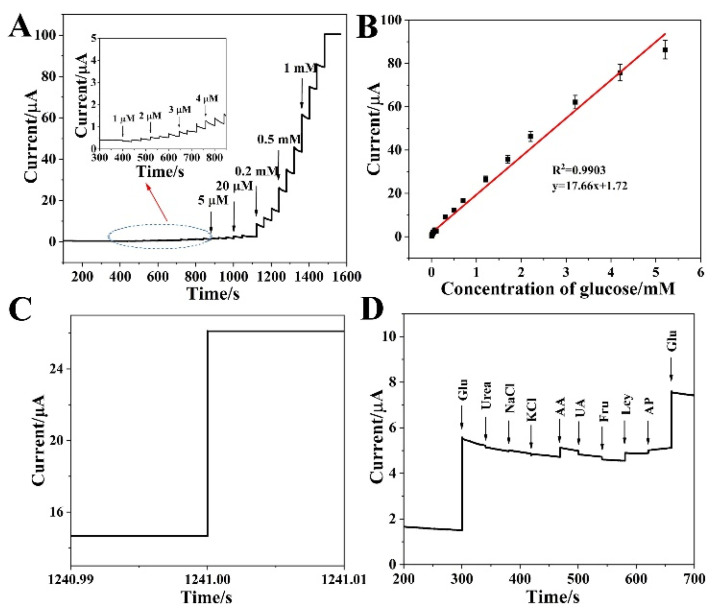
(**A**) Amperometric response of the P-NiCo_2_S_4_/ITO with successive addition of glucose at 0.4 V in 0.2 M NaOH (inset: the current response of electrode towards adding glucose from 1 to 4 μM). (**B**) The corresponding calibration curve of the P-NiCo_2_S_4_/ITO electrode to successive additions of glucose ranging from 1 µM to 5.2 mM. (**C**) The response time of the P-NiCo_2_S_4_/ITO electrode. (**D**) Amperometric response of the P-NiCo_2_S_4_/ITO with successive addition of glucose (1 mM), interfering species (0.1 mM), and the second addition of glucose (1 mM) in 0.2 M NaOH.

**Figure 6 biosensors-12-00823-f006:**
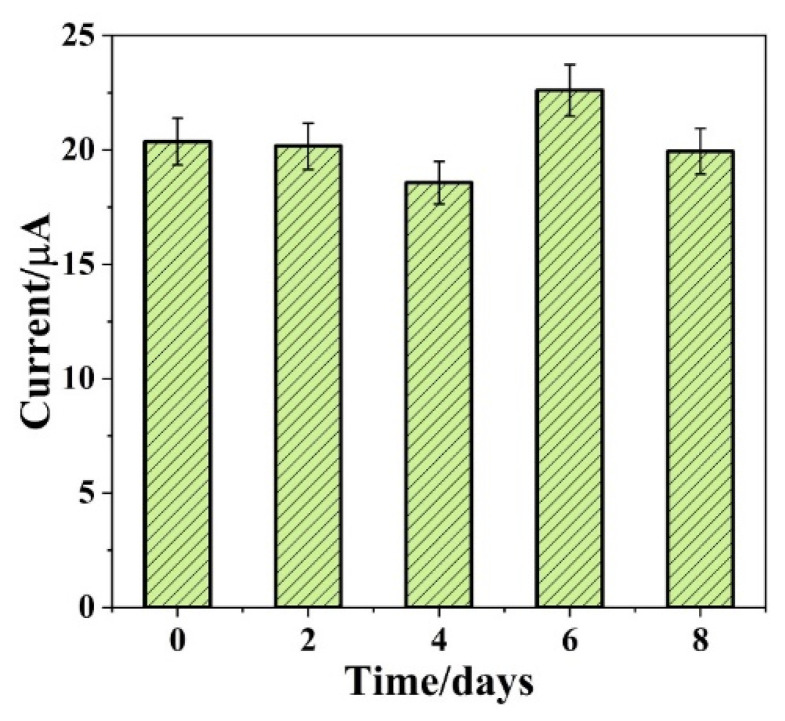
The stability of P-NiCo_2_S_4_/ITO stored at room temperature over 7 days in the presence of 1 mM glucose.

**Table 1 biosensors-12-00823-t001:** Performance comparison of P-NiCo_2_S_4_/ITO sensor toward glucose oxidation with some similar electrodes.

Electrode Materials	Sensitivity (μA mM^−1^ cm^−2^)	Linear Range (mM)	LOD (μM)	Ref.
Co_3_O_4_/NiCo_2_O_4_	304	0.01–3.52	0.384	[33]
NiCo_2_S_4_/Pt	5.14	0.001–0.664	1.2	[34]
CoNi_2_S_4_@NCF	6.675 54.82	0.5–12.5 12.5–30	NR	[35]
Ni_5_P_4_	149.6	0.002–5.3	0.7	[36]
NiS-rGO	NR	0.05–1.7	10	[37]
CoP/GCE	116.8	0.5–5.5	9	[38]
**P-NiCo_2_S_4_/ITO**	**250**	**0.001–5.2**	**0.46**	**This work**

**Table 2 biosensors-12-00823-t002:** The recovery of glucose by standard addition method in the serum sample.

Sample	Added Concentration/mM	Measured Concentration/mM	Recovery/%
Serum	0.055	0.056	101.8
	0.275	0.271	98.6
	0.475	0.458	96.4

## Data Availability

Not applicable.

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
