# Peer review of "Defect Surface Engineering of Hollow NiCo2S4 Nanoprisms towards Performance-Enhanced Non-Enzymatic Glucose Oxidation"

_biosensors, 2022, doi:10.3390/bios12100823_

Round 1

Reviewer 1 Report

This study was well organized by the authors. The all characterizations have been performed in good shape. EDX analysis should be also add to results. 

If they add this study it would be better for the characterization of structures.

Some minor english and grammer problems are avaliable. If they checked again by a native speaker , it would be good for paper.

Author Response

Reviewer: 1

This study was well organized by the authors. The all characterizations have been performed in good shape.

Q1: EDX analysis should be also added to results. If they add this study it would be better for the characterization of structures.

A1: Thanks for your suggestion. The EDX analyses of the Co/Ni precursors and P-NiCo2S4 HNPs were provided as Figure S2 and S3. And Figure S3 confirmed the uniform distribution of Co, Ni, S and P elements within the hollow prisms.

Q2: Some minor English and grammar problems are available. If they checked again by a native speaker, it would be good for paper.

A2: Thanks for your suggestion. We have corrected the grammar problems of the manuscript, which was highlighted with red font.

Reviewer 2 Report

In this paper, hollow NiCo2S4 nanoprisms electrode was developed and employed for the non-enzymatic electrochemical detection of glucose. Investigated the effects of phosphorus doping on the electrochemical performances of NiCo2S4 electrodes (P-NiCo2S4). The work is quite interesting, and authors also presented well. However, there are some clarifications to be made before publication in this journal. Therefore, I recommend that a major review should be done.

 Comments

1.      Both K4[Fe(CN)6] and K3[Fe(CN)6] written as Potassium cyanate ferrite?

 2.      As the sample contains Sulfur and Phosphorous it is important to mention the applied voltage and current when measuring the SEM and TEM of the samples

 3.      Few graphs such as XRD and so on of NiCo2S4 are repeated from authors earlier paper Dalton Trans., 2021, 50, 15162–15169. Replace it with new data or remove it and cite earlier paper and compare your data.

 4.      Include the electrochemical equivalent circuit in impedance spectra for better understanding.

 5.      In conclusion, include the limitation of the proposed sensor and mention how it can be overcome in future research

 6.      There are lots of English grammar and typo mistakes. Whole manuscript need to check for the English grammatical errors.

Author Response

In this paper, hollow NiCo2S4 nanoprisms electrode was developed and employed for the non-enzymatic electrochemical detection of glucose. Investigated the effects of phosphorus doping on the electrochemical performances of NiCo2S4 electrodes (P-NiCo2S4). The work is quite interesting, and authors also presented well. However, there are some clarifications to be made before publication in this journal. Therefore, I recommend that a major revision should be done.

Q1: Both K4[Fe(CN)6] and K3[Fe(CN)6] written as Potassium cyanate ferrite?

A1: Thanks for your question. In the manuscript, K4[Fe(CN)6] has been changed to potassium hexacyanoferrate (II), while K3[Fe(CN)6] has been changed to potassium ferricyanide.

Q2: As the sample contains Sulfur and Phosphorous it is important to mention the applied voltage and current when measuring the SEM and TEM of the samples.

A2: Thanks for your suggestion. We have added the applied voltage and current in the manuscript. The morphologies and structures of the as-prepared samples were characterized by using Scanning electron microscopy (SEM, Zelss Sigma300) at 3 kV and transmission electron microscopy (TEM, FEI Talos-F200s) at a voltage of 200 kV, which have been added in section 2.2 in the revised manuscript.

Q3: Few graphs such as XRD and so on of NiCo2S4 are repeated from authors earlier paper Dalton Trans., 2021, 50, 15162-15169. Replace it with new data or remove it and cite earlier paper and compare your data.

A3: Thanks for your suggestion. We have replaced the XRD of NiCo2S4 with a new data in the manuscript. As shown in Figure 2A, the NiCo2S4 crystal structure was not distorted significantly by the introduction of P. Our earlier paper Dalton Trans., 2021, 50, 15162-15169 has been cited as Ref. 22.

Additional:

Obviously, all the diffraction peaks can well correspond to the cubic NiCo2S4 phase (JCPDS Card. No. 20-0782) [13,22-23]. The result indicated that the NiCo2S4 crystal structure was not distorted significantly by the introduction of P [13,23].

Additional reference:

[23] Gu, H.; Fan, W.; Liu, T. Phosphorus-Doped NiCo2S4 Nanocrystals Grown on Electrospun Carbon Nanofibers as Ultra-Efficient Electrocatalysts for the Hydrogen Evolution Reaction. Nanoscale Horiz. 2017, 2, 277-283.

Q4: Include the electrochemical equivalent circuit in impedance spectra for better understanding.

A4: Thanks for your suggestion. We have added the electrochemical equivalent circuit in Figure 4D. 

Q5: In conclusion, include the limitation of the proposed sensor and mention how it can be overcome in future research.

A5: Thanks for your question. The reported nanomaterials-based electrodes have been mostly applied in strong alkaline conditions for non-enzymatic glucose sensor so far. However, the electrochemical characteristics in the weak alkaline or even neutral environment close to human body fluid are rarely reported. Maybe before the test, a negative potential is first employed to reduce the water in situ, the hydrogen gets released resulting in the formation of OH-. Thus it produces an alkaline microenvironment, so as to realize glucose detection in a neutral environment. This will be further studied in the future. We have added the contents in conclusion section.

Q6: There are lots of English grammar and typo mistakes. Whole manuscript need to check for the English grammatical errors.

A6: Thanks for your suggestion. We have corrected the grammar and typo mistakes in the revised manuscript, which have been highlighted with red font.

Reviewer 3 Report

In this report, the authors have developed hollow NiCo2S4 nanoprisms  for detection of glucose using non-enzymatic glucose oxidation. For the biosensor construction, they have synthesized phosphorous doped hollow NiCo2S4 nanoprisms nanostructure which can specifically oxidize glucose. Results showed that the sensors show a good electrochemical performance on glucose detection with linear range of 0.001 to 5.2 mM and low detection limit 0.46 µM. In addition, they also showed the sensing performances in human serum without compromising any significant change. The novelty of this present work and the reason of its low LOD, is mainly concerned on the electrochemical enhanced sensing. However, according to my understanding, the hypothesis behind the mechanism and its supporting experimental evidences need to be elaborately explained before considering its publication in the Biosensors. The authors are encouraged to reinforce the role of phosphorus in the electrochemical enhanced sensing mechanism and provide the direct evidence if possible because the work publication “Dalton Trans., 2021, 50, 15162” by Xiaojun Chen et al already explained the role of hollow prism-like NiCo2S4 electrocatalyst for glucose detection. Over all, the paper is well organized and the authors can able to describe successfully that the method have some potential advantages. On the basis of the work design and the data presentation, the work should be published after major revision by addressing the below suggestions

There are few points listed below as suggestion:

1.       Why doping of NiCo2S4 nanoprisms only with Phosphorous, why not others like Nitrogen, Boron, etc?

2.       Amount of p doping might change electrochemical activity. Need to investigate it?

3.       During the synthesis, authors prepared P-NiCo2S4 with 0.1 g, 0.2 g and 0.4 g NaH2PO2 nad concluded that he one prepared with 0.2 g NaH2PO2 is the best based on amperometric current responses.  However, authors need to explain what resulted in enhanced electrochemical behaviour of 0.2 and what decreased the electrochemical activity of 0.4 and 0.1 g?

4.       Also authors need to determine the exact value of p doped amount in t the final P-NiCo2S4  product rather than mentioning the amount of precusor used.

5.       Authors mentioned the measurement of Surface area allowing Brunauer-Emmett-Teller (BET) iso- 101 therms was carried out by monitoring N2 adsorption/desorption using a NOVA 2000 sur- 102 face area analyzer (Quantachrome) at 77 K in apparatus section. However data is not shown and discussed. They need to include the data and explain the relation of surface area of P-NiCo2S4 with 0.1 g, 0.2 g and 0.4 g NaH2PO2 with detection

6.       Why 0.2 M NaOH solution is used in detection experiments for determining calibration curve? Real sample would not work at this pH.

7.       The slope of the glucose calibration curve performed in 0.2 M NaOH solution is 17.6 and the one performed in serum is 4.6. How do authors support this change in slope of calibration curves

8.       The source of the human Serum samples need to be determined. In the recovery experiments glucose added and detected was almost same. Does the Serum samples used doesn’t have any glucose concentration in it?

9.       For the electrode preparation authors mixed P-NiCo2S4 HNPs with EtOH and coated on to the ITO electrode. No binder was used or annealing was done for attachment. How can be material be attached to the electrode?

10.   In conclusion authors mentioned “P can help to increase electronic conductivity  and rich binary electroactive sites and boost surface electroactivity”. Which data explains binary electroactive sites and boost surface electroactivity?

Author Response

In this report, the authors have developed hollow NiCo2S4 nanoprisms for detection of glucose using non-enzymatic glucose oxidation. For the biosensor construction, they have synthesized phosphorous doped hollow NiCo2S4 nanoprisms nanostructure which can specifically oxidize glucose. Results showed that the sensors show a good electrochemical performance on glucose detection with linear range of 0.001 to 5.2 mM and low detection limit 0.46 µM. In addition, they also showed the sensing performances in human serum without compromising any significant change. The novelty of this present work and the reason of its low LOD, is mainly concerned on the electrochemical enhanced sensing. However, according to my understanding, the hypothesis behind the mechanism and its supporting experimental evidences need to be elaborately explained before considering its publication in the Biosensors. The authors are encouraged to reinforce the role of phosphorus in the electrochemical enhanced sensing mechanism and provide the direct evidence if possible because the work publication “Dalton Trans., 2021, 50, 15162” by Xiaojun Chen et al already explained the role of hollow prism-like NiCo2S4 electrocatalyst for glucose detection. Over all, the paper is well organized and the authors can able to describe successfully that the method have some potential advantages. On the basis of the work design and the data presentation, the work should be published after major revision by addressing the below suggestions. There are few points listed below as suggestion:

Q1: Why doping of NiCo2S4 nanoprisms only with Phosphorous, why not others like Nitrogen, Boron, etc?

A1: Thanks for your question. We choose Phosphorous (P) doping because the atomic radius and electronegativity of the P atom are close to those of the S atom. Introducing the P element could form a new bond with another element, and thus results in lattice distortion. It is therefore expected that doping the P element into NiCo2S4 would provide some new active sites and reduce the strain during redox reactions, and hence improve the overall electrochemical performances (Phys. Rev. B: Condens. Matter Mater. Phys., 2015, 91, 123-128; Electrochim. Acta 2018, 259, 955-961; Adv. Funct. Mater. 2017, 27, 1605784; Dalton Trans. 2018, 47, 8771-8778; Nanoscale Horiz. 2017, 2, 277-283).

Q2: Amount of p doping might change electrochemical activity. Need to investigate it?

A2: Thanks for your suggestion. We have investigated the effect of amount of p doping on electrochemical activity by the amperometric current responses and electrochemical impedance spectroscopy (EIS). As shown in Figure 4C and 4D, 0.2 g NaH2PO2 was the optimal admixing quantity in the following experiments, and the corresponding P-NiCo2S4 exhibited the best electrochemical performance.

Q3: During the synthesis, authors prepared P-NiCo2S4 with 0.1 g, 0.2 g and 0.4 g NaH2PO2 and concluded that the one prepared with 0.2 g NaH2PO2 is the best based on amperometric current responses. However, authors need to explain what resulted in enhanced electrochemical behavior of 0.2 and what decreased the electrochemical activity of 0.4 and 0.1 g?

A3: Thanks for your suggestion. P doping content has an effect on the electrical conductivity and electrocatalytic activity of materials. High content of P doping can significantly improve the fixation site and catalytic site of polysulfide, and thus promote the catalytic activity of materials. However, excessive P doping will seriously affect the structural integrity of the materials, which significantly reduces the electrical conductivity, as shown in Figure 4D (Energy Environ. Sci. 2013, 6, 2839-2855; Adv. Funct. Mater. 2022, 32, 2107166; Dalton Trans. 2018, 47, 8771-8778). We have added the corresponding description and references in the manuscript.

Q4: Also authors need to determine the exact value of p doped amount in the final P-NiCo2S4 product rather than mentioning the amount of precursor used.

A4: Thanks for your suggestion. We have determined the exact value of p doped amount by XPS. And the XPS result suggested a P content of about 3.8 at%. And we have added in the manuscript.

Q5: Authors mentioned the measurement of Surface area allowing Brunauer-Emmett-Teller (BET) isotherms was carried out by monitoring N2 adsorption/desorption using a NOVA 2000 sur-102 face area analyzer (Quantachrome) at 77 K in apparatus section. However data is not shown and discussed. They need to include the data and explain the relation of surface area of P-NiCo2S4 with 0.1 g, 0.2 g and 0.4 g NaH2PO2 with detection.

A5: Thanks for your suggestion. We have added the measurement of surface area of P-NiCo2S4 with 0.1 g, 0.2 g and 0.4 g NaH2PO2 in the supporting information. As shown in Figure S4, the BET surface areas of P-NiCo2S4 with 0.2 g and 0.4 g NaH2PO2 were 53.679 and 56.725 m2 g-1, respectively, which were slightly higher than that of P-NiCo2S4 with 0.1 g NaH2PO2 (43.558 m2 g-1). However, compared with pure NiCo2S4, P doping can improve the specific surface area and electrochemical activity. The corresponding contents have been added in the manuscript.

Q6: Why 0.2 M NaOH solution is used in detection experiments for determining calibration curve? Real sample would not work at this pH.

A6: Thanks for your question. As shown in Figure 4A, the utmost response towards glucose was observed in 0.2 M NaOH solution. High-alkaline conditions can generate higher oxidation state species (Ni2+/Ni3+ and Co3+/Co4+), which provide the maximal glucose oxidation responses at P-NiCo2S4/ITO (ACS Sustainable Chem. Eng. 2018, 6, 16982-16989). We have pretreated real samples before use. Human whole blood samples were centrifuged at 3000 g for 5 min to remove cells and cellular debris. Then, the sample can be used at this pH, and many previous literatures have carried out this test under this alkaline condition (Talanta 2018, 184, 136-142; ACS Appl. Mater. Interfaces 2018, 10, 39151-39160; Food Chem. 2021, 349, 129202; Sens. Actuators B Chem. 2020, 313, 128031; Sens. Actuators B Chem. 2019, 278, 126-132).

Q7: The slope of the glucose calibration curve performed in 0.2 M NaOH solution is 17.6 and the one performed in serum is 4.6. How do authors support this change in slope of calibration curves?

A7: Thanks for your question. The slope might have changed because blood serum contains many chemical components, including water, electrolytes, proteins, and other inorganic and organic substances in addition to cellular components. Therefore, it will cause some interference and affect the sensitivity. However, the corresponding calibration curve (Figure S5B) suggests that the current density of the oxidation peak possesses a good linear relationship with the concentration of glucose, indicating the feasibility of P-NiCo2S4/ITO in real sample. This phenomenon has been reported in some literatures (ACS Appl. Mater. Interfaces 2018, 10, 39151-39160; Sens. Actuators B Chem. 2020, 313, 128031; Food Chem. 2021, 349, 129202).

Q8: The source of the human Serum samples need to be determined. In the recovery experiments glucose added and detected was almost same. Does the Serum samples used doesn’t have any glucose concentration in it?

A8: Thanks for your suggestion. The serum samples were donated by Jiangsu Province Hospital, and we have added in the manuscript. In the the recovery experiments, we have subtracted the glucose concentration contained in the serum itself by testing and calculating.

Q9: For the electrode preparation authors mixed P-NiCo2S4 HNPs with EtOH and coated on to the ITO electrode. No binder was used or annealing was done for attachment. How can be material be attached to the electrode?

A9: Thanks for your question. We used the drop coating method for attachment. That is adding or coating the modified materials which are dispersed in ethanol, directly on the electrode surface, and then forming a fixed modified film on the electrode surface after solvent evaporation and drying (Analytica Chimica Acta 2015, 881, 1-23).

Q10: In conclusion authors mentioned “P can help to increase electronic conductivity and rich binary electroactive sites and boost surface electroactivity”. Which data explains binary electroactive sites and boost surface electroactivity?

A10: Thanks for your question. As shown in Figure S5, the effective surface area of the P-NiCo2S4 was calculated as 0.134 cm2 according to the Randles-Sevcik equation, which was higher than that of our previous NiCo2S4 (0.075 cm2) (Dalton Trans. 2021, 50, 15162-15169). The data explains that P doping can boost surface electroactivity. Besides, we have compared the onset potentials of NiCo2S4 and P-NiCo2S4. As shown in Figure S4, the BET surface area of P-NiCo2S4 was (53.679 m2 g-1), which was higher than that of the previous NiCo2S4 (18.02 m2 g-1). Thus, P doping can increase the electroactive sites (Appl. Catal. B Environ. 2017, 200, 448-457).

Supplement

We are so sorry and forgot a support project hosting by the author Dandan Chu. So we have added in the Acknowledgements section.

Additional:

This research was funded by the National Natural Science Foundation of China (21575064) and Jiangsu Graduate Scientific Research Innovation Program (KYCX21_1075).

In addition, we have revised the Section 2 in the manuscript.

Round 2

Reviewer 2 Report

The authors have improved the manuscript quality in the revised manuscript. The manuscript can be published in the journal.

Reviewer 3 Report

Authors have made all the chnages suggested and this manuscript can be accepted in the present from without any further changes.